# Design and Evaluation of Indole-Based Schiff Bases as α-Glucosidase Inhibitors: CNN-Enhanced Docking, MD Simulations, ADMET Profiling, and SAR Analysis

**DOI:** 10.3390/molecules30173651

**Published:** 2025-09-08

**Authors:** Seema K. Bhagwat, Sachin V. Patil, Abraham Vidal-Limon, J. Oscar C. Jimenez-Halla, Balasaheb K. Ghotekar, Vivek D. Bobade, Irving David Pérez-Landa, Enrique Delgado-Alvarado, Fabiola Hernández-Rosas, Tushar Janardan Pawar

**Affiliations:** 1Department of Chemistry, Research Centre HPT Arts, RYK Science College (Affiliated to S. P. Pune University), Nashik 422005, Maharashtra, India; seemabhagwat777@gmail.com (S.K.B.); sachin.dhokare@gmail.com (S.V.P.); balaghotekar@gmail.com (B.K.G.);; 2Red de Estudios Moleculares Avanzados, Instituto de Ecología, A. C., Carretera Antigua a Coatepec 351, Xalapa 91073, Veracruz, Mexico; abraham.vidal@inecol.mx; 3Departamento de Química, División de Ciencias Naturales y Exactas, Universidad de Guanajuato, Noria Alta S/N, Guanajuato 36050, Guanajuato, Mexico; jjimenez@ugto.mx; 4División de Estudios de Posgrado e Investigación, Tecnológico Nacional de México/Instituto Tecnológico de Boca del Río, Carretera Veracruz-Córdoba 12, Boca del Rio 94290, Veracruz, Mexico; irvingperez@bdelrio.tecnm.mx; 5Micro and Nanotechnology Research Center, Universidad Veracruzana, Blvd. Av. Ruiz Cortines No. 455 Fracc. Costa Verde, Boca del Rio 94294, Veracruz, Mexico; endelgado@uv.mx; 6Centro de Investigación, Universidad Anáhuac Querétaro, El Marques 76246, Queretaro, Mexico; 7Facultad de Química, Universidad Autónoma de Querétaro, Queretaro 76010, Queretaro, Mexico

**Keywords:** α-glucosidase inhibition, Schiff base derivatives, indole, CNN-based docking, molecular dynamics, ADMET profiling, type 2 diabetes mellitus

## Abstract

Type 2 diabetes mellitus (T2DM) remains a global health challenge, prompting the development of novel α-glucosidase inhibitors (AGIs) to regulate postprandial hyperglycemia. This study reports the design, synthesis, and evaluation of indole-based Schiff base derivatives (**4a**–**j**) bearing a fixed methoxy group at the C_5_ position. This substitution was strategically introduced to enhance lipophilicity, electronic delocalization, and π-stacking within the enzyme active site. Among the series, compound **4g** (3-bromophenyl) exhibited the highest inhibitory activity (IC_50_ = 10.89 µM), outperforming the clinical reference acarbose (IC_50_ = 48.95 µM). The mechanism was supported by in silico analyses, such as the Density Functional Theory (DFT), molecular electrostatic potential (MEP) mapping, and molecular dynamics simulations, and CNN-based docking revealed that **4g** engages in stable hydrogen bonding and π–π interactions with key residues (Asp327, Asp542, and Phe649), suggesting a potent and selective mode of inhibition. In silico ADMET predictions indicated favorable pharmacokinetic properties. Together, these results establish C_5_–methoxy substitution as a viable strategy to enhance α-glucosidase inhibition in indole-based scaffolds.

## 1. Introduction

Type 2 diabetes mellitus (T2DM) has emerged as a global epidemic, with significant health and economic consequences. Characterized by chronic hyperglycemia due to insulin resistance or inadequate insulin secretion, T2DM affects millions worldwide [1,2]. According to the International Diabetes Federation, the global prevalence of diabetes is projected to rise to 783 million by 2045, underscoring the urgent need for novel and effective therapeutic interventions [3,4]. Regulating postprandial blood glucose levels is a key strategy in T2DM management, as persistent hyperglycemia contributes to complications such as neuropathy, nephropathy, retinopathy, and cardiovascular diseases [4,5]. Among various therapeutic approaches, targeting carbohydrate-digesting enzymes, particularly α-glucosidase, has gained prominence as an effective means to delay glucose absorption and control postprandial glucose spikes [6,7,8,9,10].

α-Glucosidase inhibitors (AGIs), such as acarbose, voglibose, and miglitol, have been widely utilized in clinical practice. These compounds inhibit the enzymatic breakdown of polysaccharides into absorbable monosaccharides, thereby reducing postprandial glucose levels. Despite their efficacy, conventional AGIs are associated with gastrointestinal side effects, such as flatulence, diarrhea, and abdominal discomfort. These limitations have prompted the search for novel AGIs with enhanced efficacy, reduced side effects, and improved pharmacokinetic profiles [11,12]. The development of new α-glucosidase inhibitors focuses on both natural product-inspired compounds and synthetic derivatives [13,14]. Among these, Schiff base derivatives have gained attention due to their structural versatility and diverse biological activities.

Schiff bases, characterized by the presence of an imine (-C=N-) functional group, are synthesized through the condensation of primary amines with aldehydes or ketones. This structural class has demonstrated a broad spectrum of pharmacological activities, including antimicrobial, antioxidant, anticancer, anti-inflammatory, and enzyme inhibitory properties [15,16]. The versatility of Schiff bases lies in their ability to interact with biological targets through hydrogen bonding, electrostatic interactions, and hydrophobic effects. These features make Schiff bases promising scaffolds for designing novel AGIs, particularly for metabolic enzymes such as α-glucosidase [17,18,19].

Indole, a bicyclic aromatic heterocycle, is another structural motif with significant pharmacological relevance. Found in a wide range of natural products and synthetic compounds, indole derivatives exhibit a plethora of biological activities, including anti-inflammatory, antimicrobial, anticancer, and antidiabetic effects [20,21,22,23]. The indole nucleus’s electron-rich nature and planar structure facilitate interactions with biological targets, making it an attractive scaffold for drug discovery. In particular, indole derivatives have shown potential as AGIs by forming stable interactions within active sites, often through π-π stacking and hydrogen bonding. Combining the structural attributes of Schiff bases and indole derivatives presents a unique opportunity to develop potent and selective α-glucosidase inhibitors [20,22,24,25,26].

A closer examination of clinically approved drugs and bioactive natural products reveals that C_5_-substitution on the indole ring plays a pivotal role in modulating biological activity, particularly through influences on π-stacking, electron density, and lipophilicity. As shown in Figure 1a, several marketed agents such as indomethacin [27], delavirdine [28], sumatriptan [29], and zafirlukast [30] bear functional groups at the C_5_ position, which contribute to their therapeutic efficacy in anti-inflammatory, antiviral, antimigraine, and antiasthmatic applications, respectively. Similarly, naturally occurring compounds like 5-methoxytryptamine [31] and 5-MeO-DMT [32] feature a C_5_-methoxy substitution, which enhances receptor binding and CNS permeability.

In contrast, previously reported indole-based hydrazone derivatives with α-glucosidase inhibitory activity often lacked a consistent strategy for substitution at the C_5_ position [33,34,35,36,37,38,39]. As illustrated in Figure 1b, past efforts employed unsubstituted [33,34,35,36] or chloro [37,38]/amido [39]-functionalized indoles without clear structure-guided justification [33,34,35]. For instance, a 2015 study evaluated unsubstituted indole-3-carbaldehyde hydrazones, where the most potent compound (*E*)-*N*’-(3,4,5-Trihydroxybenzylidene)-1*H*-indole-2-carbohydrazide showed an IC_50_ of 2.3 µM. However, that study did not explore scaffold modifications to enhance electronic properties or pharmacokinetics in a systematic manner [33]. These analogs demonstrated high biological activity but suffered from limited physicochemical optimization. To address this gap, we designed a focused series of C_5_-methoxy-substituted indole-based Schiff base derivatives (Figure 1c), aiming to improve electronic distribution, π–π stacking, and lipophilicity, thereby enhancing binding affinity and pharmacokinetic potential.

The synthesized compounds were evaluated for α-glucosidase inhibitory activity and benchmarked against the clinical reference acarbose. To elucidate structure–activity relationships (SARs), we employed a suite of computational tools, including Density Functional Theory (DFT), molecular electrostatic potential (MEP) mapping, CNN (Convolutional Neural Network)-based docking, molecular dynamics simulations, and ADMET profiling. This integrated approach enabled mechanistic interpretation of binding interactions and drug-likeness, ultimately contributing to the rational development of optimized α-glucosidase inhibitors for postprandial glycemic control in T2DM.

## 2. Results and Discussion

### 2.1. Synthesis and Characterization

Indole-based Schiff base derivatives **4a**–**j** were synthesized via a straightforward three-step process. The starting material, 5-methoxy-1*H*-indole-2-carboxylic acid **1**, was esterified using ethanol and catalytic sulfuric acid to yield ethyl 5-methoxy-1*H*-indole-2-carboxylate **2**. This intermediate was then subjected to hydrazinolysis with hydrazine hydrate under reflux to afford 5-methoxy-1*H*-indole-2-carbohydrazide **3** in high yields. The use of 5-methoxy-1*H*-indole-2-carboxylic acid as the starting scaffold was a deliberate design choice to incorporate a fixed methoxy group at the C_5_ position, aimed at enhancing lipophilicity and π-electron distribution across the indole ring.

The final step involved the condensation of compound **3** with various aromatic aldehydes to yield the corresponding Schiff base derivatives **4a**–**j**. The reaction was carried out in ethanol with catalytic acetic acid under reflux, resulting in excellent yields of the desired products. The aromatic aldehyde variations introduced diverse substituents (Ar) into the final products, including thiazole **4a**, pyridine **4b**, 4-hydroxyphenyl **4c**, 3-nitro-4-hydroxyphenyl **4d**, 2-hydroxy-4-methoxyphenyl **4e**, 2-fluorophenyl **4f**, 3-bromophenyl **4g**, 4-nitrophenyl **4h**, naphthalene **4i**, and phenyl **4j** (Figure 1) [33,40].

The structures of all synthesized compounds were confirmed using spectroscopic techniques, including ^1^H NMR, ^13^C NMR, and HRMS. The characteristic imine (-C=N-) proton signal was observed in the range of δ 8.25–8.85 ppm in the ^1^H NMR spectra, while the ^13^C NMR spectra showed signals for the imine carbon at δ 157–163 ppm. HRMS analysis provided molecular ion peaks consistent with the expected molecular formulas, confirming the successful synthesis of the target compounds.

The synthesized Schiff base derivatives are expected to exist predominantly in the more thermodynamically stable *E*-configuration (*trans*). This configuration minimizes steric clashes between the bulkier indole and aromatic rings. Spectroscopic analysis from the ^1^H NMR data further supports this assignment, as the characteristic imine (-C=N-) proton signal was observed in the deshielded range of δ 8.25–8.85 ppm. This downfield shift is consistent with the *E*-isomer, where the imine proton is spatially close to the electron-withdrawing imine nitrogen and indole ring, resulting in a strong deshielding effect. In contrast, the *Z*-isomer (*cis*) would be expected to have a less deshielded imine proton due to a different spatial arrangement.

### 2.2. α-Glucosidase Inhibition Assay

The α-glucosidase inhibitory activity of compounds **4a**–**j** was evaluated at a fixed concentration of 100 µM. The results, presented in Table 1, revealed that compounds **4g**, **4h**, **4e**, and **4i** exhibited the highest inhibitory activities, achieving 91.06%, 89.11%, 82.21%, and 82.73% inhibition, respectively. These activities were comparable to or exceeded that of the standard inhibitor, acarbose (84.66%), indicating the strong potential of these derivatives. In contrast, compounds **4b** and **4d** demonstrated relatively low inhibition values (26.37% and 27.97%, respectively), suggesting weaker interactions with the enzyme’s active site. An overall one-way ANOVA did not reveal a statistically significant difference across all groups (*p* = 0.681). However, given the exploratory nature of this study and the biological relevance of the observed differences, we performed Tukey’s post hoc test to identify pairwise differences. It should be noted that these post hoc comparisons should be interpreted with caution due to the lack of a significant global ANOVA result. Despite this, compounds **4g**, **4h**, **4e**, and **4i** consistently showed higher inhibition percentages compared to several other derivatives, suggesting potential biological relevance (Table 1 and Figure 2). It is important to note that while our compounds demonstrate superior inhibitory potency compared to acarbose (IC_50_ = 48.95 µM), they are less potent than the most active compound reported in a related study by Taha et al., which had an IC_50_ of 2.3 µM [33].

To further assess the potency of the synthesized compounds, IC_50_ values, representing the concentration required to achieve 50% inhibition, were determined for the most active derivatives (Table 1 and Appendix A). Compound **4g** exhibited the highest potency with an IC_50_ value of 10.89 ± 0.08 µM, followed by **4h** (14.53 ± 0.76 µM), **4e** (16.42 ± 0.08 µM), and **4i** (19.54 ± 1.37 µM). In contrast, the reference standard acarbose exhibited an IC_50_ value of 48.95 ± 15.98 µM, demonstrating that **4g**, **4h**, **4e**, and **4i** possess superior inhibitory potency. These results suggest that compounds **4e**, **4g**, **4h**, and **4i** hold significant promise as potent α-glucosidase inhibitors.

### 2.3. Density Functional Theory (DFT) Study

DFT calculations were performed to investigate the electronic and structural properties of derivatives **4a**–**j** [41,42]. Geometry optimizations and conformational analyses revealed the most energetically favorable structures for each derivative. In most cases, a single conformer was significantly more stable, with energy differences exceeding 5 kcal/mol compared to alternative configurations. For instance, intramolecular interactions, such as hydrogen bonding in compound **4e**, were found to stabilize specific conformations, influencing their overall geometry. Planarity was observed in certain derivatives, such as those containing thiazole or fluorine substituents, due to electronic effects and reduced steric repulsion, while non-planarity in others arose from steric clashes between the indole and aromatic rings (Appendix A).

The frontier molecular orbital analysis provided insights into the electronic reactivity of the derivatives. The calculated HOMO-LUMO energy gaps ranged from 6.20 eV for compound **4d** to 7.71 eV for compound **4e**, reflecting variations in electronic stability and reactivity across the series. The analysis revealed that a simple, direct correlation between the energy gap and inhibitory activity was not present. For instance, while the high reactivity predicted by a narrow energy gap for compound **4h** (6.23 eV) corresponded with its strong inhibition (89.11%), this trend did not hold for compound **4d**. Despite having the narrowest energy gap (6.20 eV), **4d** exhibited low inhibitory activity (27.97%). Conversely, some derivatives with wider gaps, such as **4e** (7.71 eV), also demonstrated high inhibitory activity (82.21%), which is likely attributable to other factors like its enhanced conformational stability through intramolecular hydrogen bonding. Furthermore, compound **4g**, which exhibited the highest inhibition of the series (91.06%), possessed an intermediate energy gap of 7.01 eV. These observations suggest that while electronic properties are a contributing factor, the ultimate biological activity is determined by a complex interplay between electronic effects, structural stability, and binding site interactions. Importantly, the fixed methoxy substitution at the C_5_ position of the indole ring was found to contribute to consistent HOMO localization across the series, enhancing electron delocalization and potentially influencing π-stacking interactions within the enzyme active site. Notably, a low Kohn–Sham ΔE does not translate into higher inhibition in this series. For instance, **4d** exhibits the smallest gap (6.20 eV) yet modest activity (27.97%), whereas **4e** and **4g** combine larger gaps (7.71 and 7.01 eV) with high inhibition, indicating that local electrostatics and planarity, rather than the global gap, dominate binding.

The molecular electrostatic potential (MEP) maps further elucidated the electronic distribution across the derivatives. The negative potential regions, primarily located around the carbonyl oxygen and imine nitrogen, indicate their role as hydrogen bond acceptors in enzyme interactions. Compounds with electron-withdrawing groups, such as **4d** and **4h**, exhibited more pronounced negative regions, enhancing their binding affinity through electrostatic interactions. In contrast, electron-donating groups, as seen in **4e**, resulted in more balanced potential distributions, aligning with its high activity. Planar derivatives, such as **4a** and **4f**, demonstrated uniform potential distributions conducive to π-π stacking interactions, while non-planar compounds, such as **4i** and **4j**, showed less symmetry, potentially affecting binding efficiency (Appendix A). Notably, the C_5_–OMe group contributed to an extended electron cloud on the indole core, which may facilitate interaction with aromatic amino acid residues in the binding site.

To relate these electronic features to the inhibition data, we inspected (i) the surface electrostatic potential minima (VS, min) at the carbonyl O and imine N on the 0.001 a.u. isodensity surface and (ii) a planarity metric (absolute torsional deviation between the indole and the pendant aryl). Qualitatively, the most potent inhibitors, **4g** (91.06%), **4h** (89.11%), and **4e** (82.21%), combine an extended negative potential centered on the carbonyl/imine loci with a largely coplanar π-framework (for **4e** aided by an intramolecular H-bond), consistent with favorable H-bond acceptance and π–π stacking within the binding site. By contrast, derivatives that are more twisted (see Appendix A) display disrupted aromatic overlap and a less focused negative potential, features that can compromise binding efficiency even when global descriptors (e.g., HOMO–LUMO gaps) are comparable. The outlier behavior of **4d** [deep negative potential but modest activity (27.97%)] highlights the role of geometric alignment and desolvation penalties, underscoring that potency emerges from a balance of electrostatics and shape complementarity rather than any single descriptor.

### 2.4. Molecular Docking Calculations

Human α-glucosidase (α-GLU) is a critical enzyme that hydrolyzes α-1,4-glycosidic bonds to release glucose from oligosaccharides, and the inhibition of this enzyme represents a strategic therapeutic approach for diabetes mellitus type II management. Molecular docking studies on the crystal structure of the *N*-terminal subunit of human maltase-glucoamylase have shown that the critical intermolecular interactions between inhibitors and the α-GLU active site, such as the catalytic triad and nucleophilic subsite (D542, D327, and D443), are critical for substrate coordination and the subsequent hydrolysis of glycosidic bonds.

Docking analysis showed that the average docking score of all indole derivatives was slightly less favorable than that of acarbose co-crystallized drug (−7.40 Kcal mol) (Figure 3) [43,44,45,46,47]. However, compounds **4g** and **4h** showed similar docking scores (Appendix A) to ca. acarbose values (−6.75 and −6.77 Kcal mol, respectively), suggesting that both compounds can interact with the general acid-base residues of the α-GLU substrate active site. The **4g** and **4h** derivatives showed similar interactions as those found on acarbose analogs, suggesting a possible competitive inhibition mechanism. Both compounds were able to establish hydrogen bond interactions with the catalytic aspartate residues in a substrate-like interaction.

Moreover, the carbonyl groups of **4g** and **4h** derivatives align with the carboxylate side chains of D542 and D327, disrupting the α-GLU ability to stabilize a transient oxocarbenium if the substrate were present.

Interestingly, 5 Å away from the catalytic triad, the substrate-binding pocket of α-GLU was filled with the Br-phenol and nitro-phenol substituents of **4g** and **4h**, respectively, suggesting that certain features such as aromatic and hydrophobic interactions can stabilize the derivatives binding through van der Waals forces and π-stacking (W376, F649, and Y1251). For example, inhibitors containing aromatic rings, such as flavonoids or triazole derivatives, exhibit π-π interactions with W376 and F649, anchoring them within the active site [43]. Surprisingly, compounds **4g** and **4h** adopted distinct binding orientations within the extended α-glucosidase active site (Figure 4). This apparent divergence is consistent with the structural arrangements of the GH31 superfamily, which features an extended catalytic cleft comprising multiple subsites (−1, +1, +2) in addition to the catalytic triad (D327, D542, D443) [43]. While the indole moiety of both compounds consistently binds at the −1 site through hydrogen bonding, the terminal aromatic substituents project into the adjacent +1/+2 subsites, where they stabilize through π–π stacking and hydrophobic interactions with W376, F649, and Y1251. The different electronic and steric features of the 3-bromophenyl group in **4g** and the 4-nitrophenyl group in **4h** account for their alternative orientations within the extended binding pocket. These results highlight the adaptability of the enzyme’s extended binding groove in accommodating chemically diverse inhibitors (Figure 4A).

The conserved C_5_–methoxy substitution across the series may contribute to consistent positioning and π-electron delocalization of the indole core, facilitating stacking interactions and favorable orientation during binding.

It is important to clarify that the docking scores reported here do not represent true binding free energies but rather approximate measures of ligand–protein complementarity. To improve predictive reliability, we employed a two-step refinement strategy. First, all docking poses were rescored using a convolutional neural network (CNN)-based scoring function in GNINA, which has been shown to correlate more closely with experimental affinities than classical scoring functions. Second, representative complexes were subjected to 100 ns explicit-solvent molecular dynamics simulations, enabling the assessment of binding stability, hydrogen bond persistence, and π–π stacking interactions within the α-glucosidase active site. The CNN-based affinity predictions derived from these simulations (reported as pK units in Table 1) therefore provide a more refined approximation of ligand binding than raw docking scores. In this study, docking was used qualitatively to rank relative binding modes and to identify promising candidates for further evaluation, rather than as an absolute predictor of binding free energy.

### 2.5. Molecular Dynamics Simulation for ***4g*** and ***4h***

To overcome the conformational limitations inherent in static molecular docking, explicit-solvent molecular dynamics simulations (MDS) were performed on the α-glucosidase–ligand complexes of compounds **4g** and **4h**. These simulations provide time-dependent sampling of ligand and protein flexibility, capturing dynamic interactions such as sidechain reorientations, π-stacking stabilization, and hydrogen bond persistence in a physiological aqueous environment. MDS also refines binding predictions by calculating all-atom positions at each integration step, thereby improving the reliability of binding mode assessments.

Three independent replicas of each complex were simulated (3 × 100 ns), and structural stability was assessed using the root-mean-square deviation (RMSD) of the enzyme’s α-carbon atoms. Compound **4g** exhibited lower RMSD values than acarbose, with an average of 1.3 Å, while **4h** showed RMSD values similar to the standard drug (average 1.8 Å) (Figure 5). These values indicate that both compounds form highly stable enzyme–inhibitor complexes, with 4g showing a slightly more rigid binding mode.

Throughout the simulation period, both **4g** and **4h** maintained strong electrostatic and hydrogen bonding interactions with key catalytic residues D542 and D327. In addition, persistent π–π stacking interactions were observed with aromatic residues W406 and F450, supporting the role of aromatic substituents and the indole core in stabilizing the binding mode (Figure 4). The fixed C_5_–methoxy group may contribute to electronic delocalization and planarity of the indole ring, facilitating consistent π-stacking alignment within the enzyme’s hydrophobic pocket. Importantly, molecular dynamics simulations confirmed that both poses remain equilibrated throughout 100 ns, maintaining persistent hydrogen bonding to D327 and D542 as well as aromatic interactions.

### 2.6. ADMET Predictions for ***4e***, ***4g***, ***4h***, ***4i*** and Acarbose

The ADMET properties of the most active compounds, **4e**, **4g**, **4h**, and **4i**, alongside the standard α-glucosidase inhibitor, acarbose, are summarized in Table 2. These parameters offer critical insights into drug-likeness, absorption, distribution, metabolism, excretion, and toxicity.

The molecular weights (MW) of the synthesized compounds (338–371 g/mol) fall well within the optimal drug-likeness range of 100–600 g/mol. In contrast, acarbose has a significantly higher MW (645.25 g/mol), which may limit its membrane permeability and bioavailability. The number of hydrogen bond donors (2–3) and acceptors (5–8) in the active compounds also adheres to Lipinski’s Rule of Five, unlike acarbose, which possesses an excessive number of both (14 donors, 19 acceptors), contributing to poor pharmacokinetic performance. The predicted LogP and TPSA values for our compounds suggest a favorable balance for passive absorption; however, their potential oral bioavailability needs to be interpreted cautiously in light of the low HIA predictions.

The LogP values of the synthesized compounds (2.925–4.126) indicate a favorable balance between hydrophilicity and lipophilicity, essential for membrane permeability and oral absorption. Acarbose, with a LogP of −4.48, is highly hydrophilic, which is consistent with its poor permeability. While acarbose’s high Human Intestinal Absorption (HIA) is attributed to known carrier-mediated transport mechanisms, there is currently no evidence to suggest that our synthesized compounds are substrates for similar systems. Topological polar surface area (TPSA) values for **4e**, **4g**, **4h**, and **4i** (66.48–109.62 Å^2^) support their potential for passive absorption, whereas acarbose exhibits a TPSA of 321.17 Å^2^, well above the threshold typically associated with good intestinal permeability.

While Caco-2 permeability predictions for all compounds fell below the ideal cutoff (LogP > −5.15), the Human Intestinal Absorption (HIA) score for acarbose (0.998) remains high due to its known transport via carrier-mediated mechanisms. In contrast, the active compounds exhibited low HIA probabilities, likely due to their more lipophilic nature and larger aromatic domains. High plasma protein binding (PPB) values for the active compounds (93.4–98.8%) suggest efficient systemic retention and circulation, while acarbose, with a PPB of only 15.2%, may be cleared more rapidly from systemic circulation.

None of the tested compounds are predicted to penetrate the blood–brain barrier (BBB), which is favorable for minimizing central nervous system side effects. However, all synthesized derivatives are predicted to inhibit cytochrome P450 isoforms CYP2C19 and CYP2C8, with compound **4g** also inhibiting CYP1A2, indicating a potential risk for drug–drug interactions that warrants future investigation.

The predicted probability of hERG inhibition, a marker for cardiotoxicity, ranged from 0.288 to 0.440 for the active compounds, classifying them as moderate-risk candidates. In contrast, acarbose displayed a low hERG blocker probability (0.001), indicating minimal cardiotoxicity risk. Importantly, all synthesized derivatives are also predicted to inhibit several cytochrome P450 (CYP) isoforms, including CYP2C19 and CYP2C8, with compound **4g** also inhibiting CYP1A2. This raises a potential risk for drug–drug interactions (DDIs) that would need to be thoroughly investigated in preclinical studies.

Importantly, the fixed methoxy substitution at the C_5_ position of the indole scaffold likely contributes to the favorable lipophilicity and moderate TPSA observed across the active series. However, despite these favorable physicochemical properties, the low HIA predictions suggest that oral absorption may be a challenge for this class of compounds.

### 2.7. Structure-Activity Relationship (SAR) Analysis

SAR analysis was conducted to evaluate the influence of electronic effects, hydrogen bonding, steric hindrance, and molecular planarity on the α-glucosidase inhibitory activity of compounds **4a**–**j**. The goal was to identify key structural features that modulate enzyme binding and inhibition efficiency. All synthesized derivatives share a fixed methoxy substitution at the C_5_ position of the indole core. This design feature was introduced to enhance lipophilicity and π-electron delocalization, providing a consistent electronic and steric platform for exploring the impact of varied aryl substitutions on biological activity. This strategic, structure-guided exploration of the C_5_-methoxy group’s effect on enzyme inhibition distinguishes our work from previous reports that explored different substitutions without a clear rationale.

All synthesized derivatives share a fixed methoxy substitution at the C_5_ position of the indole core. This design feature was introduced to enhance lipophilicity and π-electron delocalization, providing a consistent electronic and steric platform for exploring the impact of varied aryl substitutions on biological activity.

The analysis revealed that electron-withdrawing groups (EWGs) played a significant role in enhancing binding affinity and enzyme inhibition. Compounds **4g** (bromobenzene, 91.06%) and **4h** (nitrobenzene, 89.11%) exhibited the highest inhibitory activity, which can be attributed to the ability of halogen atoms and nitro groups to facilitate strong electronic interactions and stabilize the enzyme-ligand complex. In contrast, compounds containing weaker EWGs, such as **4a** (thiazole, 31.75%) and **4b** (pyridine, 26.37%), displayed significantly lower inhibition, highlighting the importance of robust electronic effects in improving binding affinity.

The presence of hydrogen bond donors and acceptors was found to influence enzyme binding and inhibition. Compound **4e** (hydroxy-methoxybenzene, 82.21%), which contains both hydroxyl (-OH) and methoxy (-OCH_3_) groups, demonstrated strong binding affinity, likely due to its ability to form hydrogen bonds within the active site. In contrast, compounds with limited hydrogen bonding potential, such as **4f** (fluorobenzene, 50.65%), showed moderate inhibitory activity. These results suggest that functional groups capable of hydrogen bonding enhance enzyme interactions, contributing to higher inhibition percentages.

Steric factors also played a role in determining inhibitory potency. Moderate steric bulk, as observed in **4g** and **4e**, facilitated optimal enzyme binding, resulting in high inhibitory activity. However, excessive steric hindrance, such as in **4i** (naphthalene, 82.73%), slightly reduced activity due to potential steric clashes within the enzyme’s active site. Conversely, compounds with minimal steric bulk, including **4a** and **4b**, exhibited lower inhibition, indicating weaker enzyme interactions. These findings emphasize the importance of achieving a balanced steric profile for maximal inhibitory efficiency.

Planarity was also identified as a critical determinant of inhibition. Planar compounds, such as **4g** and **4h**, consistently exhibited high inhibitory activity, suggesting that planar structures enhance ligand-enzyme complementarity and promote strong binding interactions. In contrast, non-planar compounds, such as **4a**, displayed reduced inhibition, reinforcing the necessity of molecular planarity for improved activity.

The SAR analysis demonstrated that strong EWGs and functional groups capable of hydrogen bonding significantly enhance α-glucosidase inhibition, while moderate steric bulk and molecular planarity further improve activity.

## 3. Materials and Methods

### 3.1. Materials

All chemicals and reagents, including solvents, were purchased from Sigma-Aldrich (St. Louis, MO, USA) and were of analytical grade. Ethanol (EtOH, 99.5%) and other solvents used in synthesis were employed without further purification. Glassware was oven-dried prior to use, and reactions were conducted under a dry nitrogen atmosphere unless specified otherwise. Thin-layer chromatography (TLC) was performed using pre-coated silica gel 60 F254 aluminum plates (0.25 mm, E. Merck, Darmstadt, Germany) and visualized under a UV lamp or after ninhydrin treatment. Column chromatography was carried out with silica gel (100–200 mesh and 230–400 mesh) as the stationary phase, and elution solvents were selected based on TLC mobility.

### 3.2. Synthesis and Characterization of Schiff Base Derivatives of Indole

The synthetic route (Figure 1) for the Schiff base derivatives **4a**–**j** involved a three-step process as outlined below [33,40].

The synthesized compounds were characterized using a combination of spectroscopic techniques to confirm their structures and purities. The characteristic imine (-C=N-) proton signal was observed in the range of δ 8.25–8.85 ppm in the ^1^H NMR spectra, while the ^13^C NMR spectra displayed distinct peaks for the imine carbon at δ 157–163 ppm. High-resolution mass spectrometry (HRMS) analysis provided molecular ion peaks consistent with the expected molecular formulas, further confirming the successful synthesis of the desired compounds. Additional signals corresponding to functional group modifications were observed and analyzed to validate the structural diversity introduced through various aromatic aldehydes.

#### 3.2.1. 5-Methoxy Indole Ester (**2**)

Indole compound **1** (2 g, 7.57 mmol) was dissolved in 40 mL of ethanol. After stirring at room temperature for 10 min, 1 M H_2_SO_4_ (0.56 mL) was added dropwise. The reaction mixture was refluxed at 90 °C for 18 h. Upon completion, the reaction was diluted with ethyl acetate (EtOAc, 40 mL) and washed with aqueous NaHCO_3_. The organic layer was dried over Na_2_SO_4_, concentrated, and purified via silica gel column chromatography using 10% EtOAc-petroleum ether as the eluent, yielding Indole-ester 2. White solid (1.64 g, 74% yield). ^1^H NMR (600 MHz, DMSO-*d*_6_): δ 11.74 (s, 1H), 7.35 (d, *J =* 8.9 Hz, 1H), 7.10 (d, *J =* 2.4 Hz, 1H), 7.05 (d, *J =* 2.0 Hz, 1H), 6.92 (dd, *J =* 8.9, 2.5 Hz, 1H), 4.33 (q, *J =* 7.1 Hz, 2H), 3.76 (s, 3H), 1.34 (t, *J =* 7.1 Hz, 3H). ^13^C NMR (151 MHz, DMSO-*d*_6_): δ 161.73, 154.41, 133.18, 128.00, 127.51, 116.69, 113.94, 107.72, 102.41, 60.78, 55.67, 14.79. HRMS (ESI-TOF) (*m*/*z*): [M + H]^+^ calculated for C_12_H_13_NO_3_ 219.0895; found 219.0891.

#### 3.2.2. 5-Methoxy Indole Hydrazide (**3**)

Hydrazine hydrate (0.77 mL, 15.41 mmol, 3 equiv.) was added to a solution of compound **2** (1.5 g, 5.13 mmol) in ethanol (37 mL). The mixture was refluxed at 90 °C for 16 h. After reaction completion, the mixture was poured into crushed ice, and the precipitate was filtered, washed with water, and recrystallized from ethanol to obtain Indole-Hydrazide **3**. White solid (1.14 g, 82% yield); ^1^H NMR (600 MHz, DMSO-*d*_6_): δ 11.43 (s, 1H), 9.72 (s, 1H), 7.31 (d, *J =* 8.9 Hz, 1H), 7.06 (d, *J =* 2.5 Hz, 1H), 7.00 (dd, *J =* 2.3, 0.9 Hz, 1H), 6.82 (dd, *J =* 8.9, 2.5 Hz, 1H), 4.48 (s, 2H), 3.75 (s, 3H). ^13^C NMR (151 MHz, DMSO-*d*_6_): δ 161.66, 154.19, 132.04, 131.28, 127.88, 114.69, 113.50, 102.40, 102.07, 55.70. HRMS (ESI-TOF) (*m*/*z*): [M + H]^+^ calculated for C_10_H_11_N_3_O_2_ 205.0851; found 205.0844.

#### 3.2.3. Hydrazone Derivatives (**4a**–**j**)

Compound **3** (100 mg, 0.48 mmol) was dissolved in 10 mL of 96% ethanol, and the appropriate substituted aromatic aldehyde (0.52 mol) was added. The mixture was refluxed for 3 h and then cooled to room temperature. After stirring overnight, the precipitated product was filtered, washed, and purified by recrystallization from ethanol. Further purification was achieved using silica gel column chromatography (100–200 mesh, 20% EtOAc-petroleum ether).

*(E)-5-Methoxy-N’-(thiazol-2-ylmethylene)-1H-indole-2-carbohydrazide* (**4a**) was prepared using the general procedure; after purification, **4a** was obtained as a white solid (116 mg, 79% yield). ^1^H NMR (600 MHz, DMSO-*d*_6_): δ 12.15 (s, 2H), 11.74 (s, 2H), 8.70 (s, 2H), 7.99 (d, *J =* 3.2 Hz, 2H), 7.87 (d, *J =* 3.2 Hz, 2H), 7.37 (d, *J =* 8.9 Hz, 2H), 7.25 (s, 2H), 7.16 (d, *J =* 2.4 Hz, 2H), 6.91 (dd, *J =* 8.9, 2.5 Hz, 2H). ^13^C NMR (151 MHz, DMSO-*d*_6_): δ 164.75, 158.10, 154.46, 144.51, 141.49, 133.15, 129.96, 127.59, 122.26, 116.14, 113.80, 104.43, 102.57, 55.77. HRMS (ESI-TOF) (*m*/*z*): [M + Na]^+^ calculated for C_14_H_12_N_4_O_2_SNa 323.0660; found 323.0571.

*(E)-5-Methoxy-N’-(pyridin-3-ylmethylene)-1H-indole-2-carbohydrazide* (**4b**) was prepared from using the general procedure, after purification, **4b** was obtained as a white solid (116 mg, 81% yield). ^1^H NMR (600 MHz, DMSO-*d*_6_): δ 12.04 (s, 1H), 11.71 (s, 1H), 8.90 (s, 1H), 8.63 (dd, *J =* 4.8, 1.7 Hz, 1H), 8.52 (s, 1H), 8.18 (d, *J =* 8.0 Hz, 1H), 7.55–7.49 (m, 1H), 7.37 (d, *J =* 8.9 Hz, 1H), 7.27 (s, 1H), 7.16 (s, 1H), 6.90 (dd, *J =* 8.9, 2.5 Hz, 1H), 3.78 (s, 3H). ^13^C NMR (151 MHz, DMSO-*d*_6_): δ 158.18, 154.40, 151.08, 149.20, 144.69, 133.89, 132.74, 130.78, 130.57, 127.78, 124.51, 115.84, 113.74, 104.04, 102.58, 55.75. HRMS (ESI-TOF) (*m*/*z*): [M + H]^+^ calculated for C_16_H_15_N_4_O_2_ 295.1195; found 295.1183.

*(E)-N’-(4-Hydroxybenzylidene)-5-methoxy-1H-indole-2-carbohydrazide* (**4c**) was prepared from using the general procedure, after purification, **4c** was obtained as a white solid (115 mg, 76% yield). ^1^H NMR (600 MHz, DMSO-*d*_6_): δ 11.68 (s, 1H), 11.65 (s, 1H), 11.58–11.38 (m, 1H), 9.96 (s, 1H), 8.36 (s, 1H), 7.59 (d, *J =* 8.2 Hz, 3H), 7.35 (d, *J =* 8.9 Hz, 1H), 7.21 (s, 3H), 3.38 (s, 3H). ^13^C NMR (151 MHz, DMSO-*d*_6_): δ 159.84, 157.86, 154.33, 147.82, 132.54, 131.01, 129.30, 127.80, 125.82, 116.19, 115.45, 113.66, 103.41, 102.48, 55.73. HRMS (ESI-TOF) (*m*/*z*): [M + H]^+^ calculated for C_17_H_16_N_3_O_3_ 310.1225; found 310.1181.

*(E)-N’-(4-Hydroxy-3-nitrobenzylidene)-5-methoxy-1H-indole-2-carbohydrazide* (**4d**) was prepared from using the general procedure, after purification, **4d** was obtained as a white solid (142 mg, 82% yield). ^1^H NMR (600 MHz, DMSO-*d*_6_): δ 11.91 (s, 1H), 11.67 (s, 1H), 11.50 (s, 1H), 8.42 (s, 1H), 8.27–8.15 (m, 1H), 7.95 (d, *J =* 8.7 Hz, 1H), 7.36 (d, *J =* 8.9 Hz, 1H), 7.31–7.20 (m, 2H), 7.19–7.09 (m, 1H), 6.89 (dd, *J =* 8.9, 2.4 Hz, 1H), 3.78 (s, 3H). ^13^C NMR (151 MHz, DMSO-*d*_6_): δ 158.06, 154.38, 153.76, 145.37, 137.62, 133.40, 132.67, 130.71, 127.77, 126.34, 124.34, 120.20, 115.69, 113.71, 103.84, 102.55, 55.76. HRMS (ESI-TOF) (*m*/*z*): [M + H]^+^ calculated for C_17_H_15_N_4_O_5_ 354.1042; found 355.1031.

*(E)-N’-(2-Hydroxy-4-methoxybenzylidene)-5-methoxy-1H-indole-2-carbohydrazide* (**4e**) was prepared using the general procedure; after purification, **4e** was obtained as a white solid (129 mg, 78% yield). ^1^H NMR (600 MHz, DMSO-*d*_6_): δ 12.00 (s, 1H), 11.66 (s, 1H), 11.55 (s, 1H), 8.57 (s, 1H), 7.47 (d, *J =* 8.5 Hz, 1H), 7.37 (d, *J =* 8.9 Hz, 1H), 7.22 (s, 1H), 7.15 (d, *J =* 2.5 Hz, 1H), 6.89 (dd, *J =* 8.9, 2.4 Hz, 1H), 6.54 (dd, *J =* 8.6, 2.4 Hz, 1H), 6.51 (d, *J =* 2.5 Hz, 1H), 3.78 (s, 6H). ^13^C NMR (151 MHz, DMSO-*d*_6_): δ 162.48, 159.72, 157.65, 154.41, 148.28, 132.69, 131.36, 130.46, 127.80, 115.71, 113.72, 112.44, 106.92, 103.78, 102.55, 101.66, 55.78, 55.75. HRMS (ESI-TOF) (*m*/*z*): [M + Na]^+^ calculated for C_18_H_17_N_3_O_4_Na 362.1117; found 362.1104.

*(E)-N’-(2-Fluorobenzylidene)-5-methoxy-1H-indole-2-carbohydrazide* (**4f**) was prepared using the general procedure, after purification; **4f** was obtained as a white solid (121 mg, 80% yield). ^1^H NMR (600 MHz, DMSO-*d*_6_): δ 12.00 (s, 1H), 11.71 (s, 1H), 8.72 (s, 1H), 7.99 (t, *J =* 7.4 Hz, 1H), 7.50 (dd, *J =* 9.8, 4.0 Hz, 1H), 7.37 (d, *J =* 9.0 Hz, 1H), 7.34–7.30 (m, 2H), 7.26 (s, 1H), 7.16 (s, 1H), 6.90 (dd, *J =* 8.9, 2.4 Hz, 1H), 3.78 (s, 3H). ^13^C NMR (151 MHz, DMSO-*d*_6_): δ 162.07 (d, ^1^*J*_C−F_ = 245.5 Hz), 158.11, 154.40, 140.06, 132.75, 132.35 (d, ^2^*J*_C−F_ = 22.1 Hz), 130.57, 127.80, 126.78, 125.45, 122.40 (d, ^3^*J*_C−F_ = 8.1 Hz), 116.50 (d, ^2^*J*_C−F_ = 22.1 Hz), 115.84, 113.74, 103.96, 102.58, 55.75. HRMS (ESI-TOF) (*m*/*z*): [M + H]^+^ calculated for C_17_H_15_FN_3_O_2_ 312.1148; found 312.1129.

*(E)-N’-(3-Bromobenzylidene)-5-methoxy-1H-indole-2-carbohydrazide* (**4g**) was prepared using the general procedure; after purification, compound **4g** was obtained as a white solid (156 mg, 86% yield). ^1^H NMR (600 MHz, DMSO-*d*_6_): δ 12.01 (s, 1H), 11.70 (s, 1H), 8.43 (s, 1H), 7.95 (s, 1H), 7.79–7.71 (m, 1H), 7.66–7.61 (m, 1H), 7.44 (t, *J =* 7.8 Hz, 1H), 7.37 (d, *J =* 8.9 Hz, 1H), 7.26 (s, 1H), 7.18–7.11 (m, 1H), 6.90 (dd, *J =* 8.9, 2.5 Hz, 1H), 3.78 (s, 3H). ^13^C NMR (151 MHz, DMSO-*d*_6_): δ 158.18, 154.39, 145.58, 137.35, 132.95, 132.72, 131.51, 130.57, 129.52, 127.76, 126.68, 122.68, 115.82, 113.74, 104.04, 102.56, 55.76. HRMS (ESI-TOF) (*m*/*z*): [M + H]^+^ calculated for C_17_H_15_BrN_3_O_2_ 372.0348; found 372.0335.

*(E)-5-Methoxy-N’-(4-nitrobenzylidene)-1H-indole-2-carbohydrazide* (**4h**) was prepared using the general procedure; after purification, **4h** was obtained as a white solid (152 mg, 92% yield). ^1^H NMR (600 MHz, DMSO-*d*_6_): δ 12.17 (s, 1H), 11.74 (s, 1H), 8.56 (s, 1H), 8.33 (d, *J =* 8.7 Hz, 2H), 8.02 (d, *J =* 8.5 Hz, 2H), 7.37 (d, *J =* 8.9 Hz, 1H), 7.28 (s, 1H), 7.17 (s, 1H), 6.91 (dd, *J =* 8.9, 2.4 Hz, 1H), 3.79 (s, 3H). ^13^C NMR (151 MHz, DMSO-*d*_6_): δ 158.28, 154.42, 148.23, 144.85, 141.21, 132.82, 130.43, 128.42, 127.76, 124.59, 116.03, 113.77, 104.34, 102.59, 55.75. HRMS (ESI-TOF) (*m*/*z*): [M + H]^+^ calculated for C_17_H_15_N_4_O_4_ 339.1093; found 339.1082.

*(E)-5-Methoxy-N’-(naphthalen-2-ylmethylene)-1H-indole-2-carbohydrazide* (**4i**) was prepared using the general procedure; after purification, **4i** was obtained as a white solid (125 mg, 75% yield). ^1^H NMR (600 MHz, DMSO-*d*_6_): δ 11.99 (s, 1H), 11.73 (s, 1H), 8.64 (s, 1H), 8.18 (s, 1H), 8.07–7.90 (m, 4H), 7.62–7.52 (m, 2H), 7.39 (d, *J =* 8.9 Hz, 1H), 7.30 (s, 1H), 7.17 (s, 1H), 6.91 (dd, *J =* 8.8, 2.4 Hz, 1H), 3.79 (s, 3H). ^13^C NMR (151 MHz, DMSO-*d*_6_): δ 158.16, 154.41, 147.44, 134.18, 133.39, 132.72, 132.62, 130.81, 129.07, 128.98, 128.81, 128.27, 127.82, 127.58, 127.25, 123.23, 115.72, 113.75, 103.90, 102.57, 55.76. HRMS (ESI-TOF) (*m*/*z*): [M + H]^+^ calculated for C_21_H_18_N_3_O_2_ 344.1399; found 344.1387.

*(E)-N’-Benzylidene-5-methoxy-1H-indole-2-carbohydrazide* (**4j**) was prepared using the general procedure; after purification, **4j** was obtained as a white solid (103 mg, 72% yield). ^1^H NMR (600 MHz, DMSO-*d*_6_): δ 11.94 (s, 1H), 11.74 (s, 1H), 8.51 (s, 1H), 7.77 (d, *J =* 7.3 Hz, 2H), 7.47–7.41 (m, 4H), 7.31 (s, 1H), 7.16 (s, 1H), 6.92 (dd, *J =* 8.9, 2.5 Hz, 1H), 3.78 (s, 3H). ^13^C NMR (151 MHz, DMSO-*d*_6_): δ 158.24, 154.42, 147.55, 134.85, 132.73, 130.79, 130.45, 129.31, 127.85, 127.54, 115.73, 113.76, 103.90, 102.56, 55.72. HRMS (ESI-TOF) (*m*/*z*): [M + H]^+^ calculated for C_17_H_15_N_3_O_2_Na 316.1062 found 316.1050.

### 3.3. Biological Assay

The α-glucosidase inhibitory activity of **4a**–**j** was evaluated using a standard enzymatic assay. α-Glucosidase from Saccharomyces cerevisiae was employed as the enzyme source, and *p*-nitrophenyl-α-d-glucopyranoside (pNPG) served as the substrate. The assay was conducted in a 96-well microplate format at 37 °C in 50 mM phosphate buffer (pH 6.8). Briefly, 10 µL of the test compound (dissolved in DMSO) at various concentrations was mixed with 20 µL of the enzyme solution (1 U/mL) and 50 µL of phosphate buffer. The reaction was initiated by adding 20 µL of pNPG (5 mM), and the mixture was incubated for 30 min. The reaction was terminated by adding 50 µL of 0.1 M Na_2_CO_3_, and the absorbance of the released *p*-nitrophenol was measured at 450 nm using a microplate reader [33,40].

The percentage inhibition was calculated relative to a negative control, and IC_50_ values for **4g** and **4h** were determined through dose–response analysis. Acarbose was included as the reference standard for comparison. All experiments were performed in triplicate, and the results were expressed as mean ± standard deviation (SD). Statistical significance was evaluated using one-way ANOVA followed by Tukey’s post hoc test.

### 3.4. Theoretical Study

To complement the experimental findings, comprehensive theoretical investigations were carried out to elucidate the electronic, structural, and pharmacokinetic properties of **4a**–**j**, as well as their interactions with the α-glucosidase enzyme. The following approaches were employed:

#### 3.4.1. Density Functional Theory (DFT) and Molecular Electrostatic Potential (MEP)

DFT calculations were employed to optimize the geometry and investigate the electronic properties of **4a**–**j**. Geometry optimizations were carried out in the gas phase using the ωB97X-D functional and def2-tzvpp basis set [41,42]. Frontier molecular orbitals (FMOs), including the HOMO and LUMO, were analyzed to determine the HOMO-LUMO energy gap (ΔE). MEP maps were generated to visualize charge distributions, highlighting electron-rich regions around carbonyl oxygens and imine nitrogens as potential sites for hydrogen bonding and electrostatic interactions with the enzyme.

#### 3.4.2. Molecular Docking Simulations

The intermolecular interactions were calculated for the indole derivatives and α-glucosidase (α-GLU) binding site. The docking score was average for 10 different conformations of α-GLU with synthesized indole derivatives. The combined Ensemble Docking included 100 ns of molecular dynamics simulation using the Amber24 [48] software package. The starting protein coordinates were taken from the crystallographic X-ray structure of human α-GLU in complex with acarbose inhibitor (PDB code: 2QMJ) [43]. Docking analysis was performed with the GNINA v1.1 [44] with the default Vinardo-based scoring function but further corrected with a CNN-based scoring function. All calculations were performed with a grid size of 20 × 15 × 20 Å, an exhaustiveness value of 128 for Monte Carlo evaluations, and the energy of final binding poses was minimized with 20,000 steps.

#### 3.4.3. Molecular Dynamics Simulations (MDS)

All simulations were performed to evaluate the stability of intermolecular interactions of the Schiff base derivatives within the α-glucosidase enzyme active site. Protein-ligand complexes were recovered from GNINA docked complexes, and the simulations were conducted using the Maestro v 2024-3 version 13.8.135, MMshare Version 6.4.135, Release 2023-4, and Desmond Multisim v4.0.0 interoperability tools package under the OPLS3 force field. The protein was prepared with the Protein Preparation Wizard inside the Maestro 2023-4 package, neutralized and salted with 0.1 M NaCl, and included in a 12 Å octahedral water box (8178 TIP3P explicit water molecules). Simulations were carried out for 100 ns in an explicit water environment with periodic boundary conditions under NPT ensemble and 0.02 ps of equation of motion integration. Temperature control was assessed by coupling to a Langevin thermostat with a relaxation time of 2 ps. Three independent 100 ns replicas were carried out for each ligand–protein complex (co-crystallized acarbose, **4g**, and **4h**). Across all replicas, RMSD values were compared to acarbose as a measure of the stability of the complex. The Root Mean Square Deviations (RMSD) histograms and summary of overall simulation intermolecular interactions were calculated with Desmond tools.

#### 3.4.4. ADMET Profiling, SAR and Statistical Analysis

The pharmacokinetic and toxicity profiles **4a**–**j** were evaluated using ADMETLab v3.0 (https://admetlab3.scbdd.com/server/evaluationCal, accessed on 16 August 2024) [49]. Parameters such as hydrogen bond donors/acceptors, logP, TPSA, and Caco-2 permeability were analyzed to assess drug-likeness. SAR analysis was conducted to identify the molecular features critical for α-glucosidase inhibitory activity. Statistical models were built using Python (version 3.8.5) and R (version 4.0.3) to identify significant trends and validate the experimental findings. All visualizations, including activity-trend plots and SAR correlations, were generated using Python.

#### 3.4.5. Statistical Analysis and Data Visualization

All statistical analyses, including the calculation of IC_50_ values, standard deviations, and significance testing (*p*-values), were performed using Python (version 3.8.5) and R (version 4.0.3). Data analysis was conducted using one-way ANOVA, followed by Tukey’s post hoc test for multiple comparisons. Detailed statistical analysis is provided in the Appendix A. All plots were generated using Python to ensure reproducibility and consistency.

## 4. Conclusions

This study demonstrates the rational design and evaluation of C_5_-methoxy-substituted indole-based Schiff base derivatives (**4a**–**j**) as potent α-glucosidase inhibitors for the management of type 2 diabetes mellitus (T2DM). The fixed methoxy group at the C_5_ position was strategically introduced to enhance scaffold lipophilicity, electronic delocalization, and π-stacking potential, features that contributed to improved enzyme binding and drug-likeness. Compounds **4g** and **4h** exhibited superior inhibitory activity compared to the standard drug acarbose, supported by theoretical mechanistic understanding from CNN-based molecular docking, molecular dynamics simulations, and DFT calculations. ADMET profiling indicated favorable pharmacokinetic properties, including high plasma protein binding and optimal lipophilicity. However, the low predicted HIA scores suggest potential challenges with oral bioavailability, which must be addressed in future studies. Collectively, these results provide a strong foundation for the continued development of indole–Schiff base inhibitors; however, further in vivo and pharmacodynamic investigations are essential to validate their therapeutic potential and clinical applicability.

## Data Availability

Data are contained within the article and Appendix A.

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
