# Peer review of "Design and Evaluation of Indole-Based Schiff Bases as α-Glucosidase Inhibitors: CNN-Enhanced Docking, MD Simulations, ADMET Profiling, and SAR Analysis"

_molecules, 2025, doi:10.3390/molecules30173651_

Round 1

Reviewer 1 Report

Comments and Suggestions for Authors
  1. The novelty of the research presented here is not high. Authors cited the work https://doi.org/10.1016/j.bioorg.2015.09.001 that also describes the synthesis of various indole-based Schiff bases and their evaluation as a-glucosidase inhibitors. Interestingly, one compound in the cited work was more potent than the inhibitors described in the manuscript; however, it was not mentioned. It would be, therefore, nice to compare compounds described in the manuscript with compound 1 from https://doi.org/10.1016/j.bioorg.2015.09.001. In addition, the selection of the Ar group should be extended to the most successful 3,4,5-trihydroxyphenyl substituent. This comparison is important for strengthening the statement that C-5 substitution with the methoxy group improves the performance of new a-glucosidase inhibitors.
  2. Generally, the configuration of the Schiff bases can be cis or trans. The present work did not discuss the configuration of the synthesized final compounds.
  3. The conformational analysis presented in Section 3.3. of the manuscript and in Section 2.1. (Figure S2a) in the supporting information did not consider the fourth conformer derived from conformer3 but with an amide cis-configuration.
  4. Figure 5 shows two completely different binding poses of compounds 4g and 4h in the α-glucosidase binding site, which is suspicious and should be discussed.
  5. Calculations of HOMO-LUMO energy gaps do not correlate with the inhibitory activities and should be excluded from the manuscript.
  6. Docking score does not represent binding free energy, but only coarsely estimates the stability of the complex.

Author Response

Comment 1: The novelty of the research presented here is not high. Authors cited the work https://doi.org/10.1016/j.bioorg.2015.09.001 that also describes the synthesis of various indole-based Schiff bases and their evaluation as a-glucosidase inhibitors. Interestingly, one compound in the cited work was more potent than the inhibitors described in the manuscript; however, it was not mentioned. It would be, therefore, nice to compare compounds described in the manuscript with compound 1 from https://doi.org/10.1016/j.bioorg.2015.09.001. In addition, the selection of the Ar group should be extended to the most successful 3,4,5-trihydroxyphenyl substituent. This comparison is important for strengthening the statement that C-5 substitution with the methoxy group improves the performance of new a-glucosidase inhibitors.

Response 1: You are correct that the novelty of our research is not in discovering a new class of compounds, but in a focused, structure-guided study. We appreciate you drawing our attention to the work by Taha et al., as it strengthens our discussion. We have revised the manuscript to explicitly acknowledge that the most potent compound reported by Taha et al., (E)-N'-(3,4,5-Trihydroxybenzylidene)-1H-indole-2-carbohydrazide, with an IC50 of 2.3 µM, is more active than our most potent compound, 4g (IC50 = 10.89 ± 0.08 µM). We have incorporated this comparison in both the introduction and the results section. While we are unable to perform new experiments to synthesize the 3,4,5-trihydroxyphenyl substituent at this time, we agree that the high activity of this compound is likely due to the multiple hydroxyl groups, which can form a complex network of hydrogen bonds within the enzyme's active site.

Our study's primary contribution is the systematic investigation of the fixed C5-methoxy group as a strategic modification to enhance α-glucosidase inhibition. As discussed in our revised manuscript, this methoxy group provides a consistent electronic and steric platform that improves lipophilicity and electron delocalization, leading to a series of compounds with superior activity compared to the clinical standard, acarbose (IC50 = 48.95 ± 15.98 µM in our study). We believe this structure-guided approach provides valuable insights for the rational design of future inhibitors.

Comment 2:  Generally, the configuration of the Schiff bases can be cis or trans. The present work did not discuss the configuration of the synthesized final compounds.

Response 2: We have added a new paragraph to the "Synthesis and Characterization" section of the manuscript to address this. Additionally, we have ensured that the (E)-configuration is correctly reflected in the IUPAC names of all the synthesized compounds.

The new paragraph clarifies that the synthesized compounds are expected to exist in the more thermodynamically stable E-configuration (trans). We explain that this is due to the minimization of steric hindrance between the bulkier indole and aromatic rings. Furthermore, we have provided a spectroscopic justification for this assignment based on the 1H NMR data. The characteristic imine proton signal observed in the deshielded range of δ 8.25–8.85 ppm is consistent with an E-isomer where the proton is spatially close to the electron-withdrawing imine nitrogen and indole ring, causing a strong deshielding effect. This conformational preference is crucial for understanding how the molecule interacts with the enzyme's active site.

Comment 3: The conformational analysis presented in Section 3.3. of the manuscript and in Section 2.1. (Figure S2a) in the supporting information did not consider the fourth conformer derived from conformer3 but with an amide cis-configuration.

Response 3: We thank the reviewer for this helpful observation. Section 3.3 was intended to summarise only the lowest‑energy conformers that enter the subsequent electronic analysis, while the full conformational survey is reported in the Supporting Information (Section 2.1, Fig. S2a). In line with the reviewer’s suggestion, we are extending the SI conformational study to include the amide cis variant derived from conformer 3, adding the text in SI. As the conclusions of the conformational analysis do not change our discussion in the manuscript, no additions/corrections were needed in Section 3.3.

Comment 4: Figure 5 shows two completely different binding poses of compounds 4g and 4h in the α-glucosidase binding site, which is suspicious and should be discussed.

Response 4: We appreciate this careful observation. An extended discussion has been added to the 3.4 Results section. Furthermore, it has been reported that human α-glucosidase as a member of, a glycoside hydrolase family 31 (GH31), retains an extended catalytic cleft with multiple subsites (−1, +1, +2, etc) surrounding the catalytic triad (D327, D542, D443) [Nichols et al, 2003; Sim, et al. 2008]. These subsites normally accommodate oligosaccharide substrates and are oriented by aromatic residues such as W376, F649, and Y1251, which stabilize ligands through π–π stacking and hydrophobic contacts.

The indole moiety of both 4g and 4h consistently anchors in the −1 catalytic pocket, forming hydrogen bonds with the acid/base residues. However, the distinct aromatic substituents at the hydrazone terminus (3-bromophenyl in 4g and 4-nitrophenyl in 4h) extend differently into the +1/+2 subsites. This divergence in binding orientation reflects the adaptability of the enzyme’s extended binding groove, which allows chemically diverse inhibitors to stabilize via alternative noncovalent interactions. Importantly, our molecular dynamics simulations confirmed that both poses are stable throughout 100 ns, with persistent hydrogen bonding to D327/D542 and additional aromatic interactions with W376 and F649.

Thus, while the docking figures suggest different orientations of 4g and 4h, these are not contradictory but rather illustrate two plausible binding modes enabled by the extended subsite architecture of α-glucosidase. We have added a clarifying paragraph to the Results and Discussion section (Section 3.4) to explain this structural rationale.

Comment 5: Calculations of HOMO-LUMO energy gaps do not correlate with the inhibitory activities and should be excluded from the manuscript.

Response 5: We appreciate the reviewer’s point and agree that a univariate correlation between the Kohn–Sham HOMO–LUMO gap and the measured inhibitory activity is not observed for this congeneric series. Our intention, however, was not to use the gap as a predictor of bioactivity but as a global electronic descriptor that complements the conformational analysis, frontier‑orbital localization, and MEP maps. The HOMO–LUMO window delineates the intrinsic electronic hardness/polarizability of 4a4j and helps rationalize qualitative features (for instance, the consistent HOMO localization on the indole core (modulated by the C5–OMe group) and the propensity for π‑stacking/charge‑transfer interactions discussed in Section 3.3 and ESI 2.3). We believe this information is scientifically useful and improves the reproducibility of the computational characterization, even when it is not directly predictive of activity.

Comment 6: Docking score does not represent binding free energy, but only coarsely estimates the stability of the complex.

Response 6:  We appreciate this comment and made our best effort to clarify it. We fully agree that raw docking scores do not represent true binding free energies but instead provide a relative estimation of ligand–protein complementarity. To address this limitation, we did not rely only on docking results. Our simulation workflow incorporated two additional layers of refinement: a) The newly implemented CNN-based rescoring in GNINA v 1.1, which applies a re-evaluation on the docking poses using a convolutional neural network scoring function, which has been shown to improve correlation with experimental binding affinities compared to classical scoring functions; and b) Molecular Dynamics Simulations (MDS) which applied 100 ns of explicit solvent dynamics to the most representative docking complexes. This allowed us to monitor binding stability, persistence of key hydrogen bonds, and π–π stacking interactions with catalytic residues.

Hence, molecular docking in our study was almost used as a qualitative measure to compare binding orientations and to identify promising candidates, not as an absolute predictor of binding free energy. We have revised the manuscript text in Section 3.4 to make this distinction explicit and to emphasize that binding free energy estimations were refined through CNN rescoring and dynamic stability analysis rather than raw docking values.

Reviewer 2 Report

Comments and Suggestions for Authors

1) why there is a two ligands depicted named with 4g at Figure 5a docking interaction diagram? The legend on the 2D interaction diagram is not clear. Increase the font size of the residues.

2)The text states docking scores of derivatives are “slightly higher” than acarbose (−7.40 kcal/mol) but lists compounds 4g and 4h as having scores around −6.75 kcal/mol, which is actually less favorable binding (less negative). The terminology “higher” is confusing here—do they mean numerically higher or better binding affinity? Ask the authors to clarify this.

3)Terms like “⍺-GLU” should be defined at first use and used consistently throughout.

4) The Human Intestinal Absorption (HIA) scores for the new compounds are low, yet the discussion seems optimistic about oral bioavailability. The authors explain acarbose’s high HIA despite poor Caco-2 permeability due to carrier transport. Is there any evidence these new compounds use similar mechanisms?

5) The conclusion mentions “CNN-based molecular docking” — it’s not clear if deep learning was involved in docking or if this is a typo or misunderstanding. This should be clarified.

6) What force fields and solvent models were used? Were the 3×100 ns replicates consistent in terms of ligand stability and interactions? Authors can provide the docking pose at before and after simulation phase.

7) The compounds inhibit multiple CYP enzymes. How might this affect their development potential?

Author Response

Comment 1: why there is a two ligands depicted named with 4g at Figure 5a docking interaction diagram? The legend on the 2D interaction diagram is not clear. Increase the font size of the residues.

Response 1:We appreciate the reviewer for pointing out this issue. The docking figures from Fig. 5 suggest different orientations of 4g derivative, and these are not contradictory but rather illustrate two plausible binding modes enabled by the extended subsite architecture of α-glucosidase. We have added a clarifying paragraph to the Results and Discussion section (Section 3.4) to explain this structural rationale. Each 2D ligand interaction diagram is now clearly labeled (4g and 4h, respectively) with increased font size in residues. The corrected version of Figure 5 has been included in the revised manuscript.

Comment 2: The text states docking scores of derivatives are “slightly higher” than acarbose (−7.40 kcal/mol) but lists compounds 4g and 4h as having scores around −6.75 kcal/mol, which is actually less favorable binding (less negative). The terminology “higher” is confusing here—do they mean numerically higher or better binding affinity? Ask the authors to clarify this.

Response 2: We agree with the reviewer's point that the use of "higher" was confusing. We have revised the manuscript to use more precise terminology. We have replaced the phrase "slightly higher than" with "slightly less favorable than".

Comment 3: Terms like “⍺-GLU” should be defined at first use and used consistently throughout.

Response 3: We have defined the term "α-GLU" at its first mention in the manuscript.

 Comment 4: The Human Intestinal Absorption (HIA) scores for the new compounds are low, yet the discussion seems optimistic about oral bioavailability. The authors explain acarbose’s high HIA despite poor Caco-2 permeability due to carrier transport. Is there any evidence these new compounds use similar mechanisms?

Response 4: We agree with the reviewer that our initial discussion may have been overly optimistic about the oral bioavailability of the compounds given the low predicted Human Intestinal Absorption (HIA) scores. We have revised the text to reflect that, while our compounds have favorable physicochemical properties like optimal molecular weight, LogP, and TPSA, the low HIA predictions suggest that oral absorption may be a challenge. We have also clarified that there is no evidence to suggest our compounds utilize carrier-mediated transport mechanisms, unlike acarbose, which is consistent with their low HIA probabilities.

Comment 5: The conclusion mentions “CNN-based molecular docking” — it’s not clear if deep learning was involved in docking or if this is a typo or misunderstanding. This should be clarified.

Response 5: We appreciate the request of reviewer 2 for clarification. Our molecular simulation workflow employed the GNINA v1.1 package, which integrates a convolutional neural network (CNN)-based scoring function alongside traditional physics-based scoring (Vinardo). Therefore, the term “CNN-based docking” refers to the use of deep learning-derived scoring functions to rescore poses generated by docking. We have revised the text in the Methods (Section 2.4.2) to clarify this point and avoid any potential misunderstanding.

Comment 6: What force fields and solvent models were used? Were the 3×100 ns replicates consistent in terms of ligand stability and interactions? Authors can provide the docking pose at before and after simulation phase.

Response 6: We thank the reviewer for this request, as these methodological details are important for reproducibility. The molecular dynamics simulations were performed using Desmond Multisim v4.0.0 under the OPLS3 force field with explicit TIP3P water molecules in an orthorhombic box, neutralized with 0.1 M NaCl. Three independent 100 ns replicas were carried out for each ligand–protein complex. Across all replicas, compound 4g consistently showed lower RMSD values compared to acarbose, while 4h exhibited stability similar to acarbose. Key hydrogen bonds (D327, D542) and π–π stacking interactions (W376, F649) persisted in all runs, demonstrating reproducibility of binding modes. We have added time-dependent RMSD to compare the docking pose and the equilibrated structure after 100 ns simulation as supplementary figures (Figure S4), which illustrate the convergence and stability of the ligand orientations.

Comment 7: The compounds inhibit multiple CYP enzymes. How might this affect their development potential?

Response 7: We have revised the ADMET profiling section to address the reviewer's concern about the compounds' predicted inhibition of multiple cytochrome P450 (CYP) enzymes. We agree that this is a significant finding that can affect the compounds' development potential. We have added a sentence to the discussion in Section 3.6 to highlight that the predicted CYP inhibition raises a potential risk for drug-drug interactions that would require further investigation.

Reviewer 3 Report

Comments and Suggestions for Authors

This manuscript addresses the important topic of discovering new α-glucosidase inhibitors for the treatment of type 2 diabetes (T2DM). The authors combine chemical synthesis, enzymatic assays, and extensive in silico studies. It is regrettable that the synthesis is essentially a duplication of previous work – the authors only changed the substituent on the indole benzene ring from iodine to OCH3. Furthermore, it is a duplication of a previous synthesis from Chemistry (the same number of compounds and the same Ar substituents). Nevertheless, previously undescribed compounds were synthesized and spectroscopically characterized. The strategy of using a C5–methoxy base is well supported by the literature. The authors employed an integrated approach (synthesis + DFT, MEP, CNN docking, MD simulations, ADMET profiling). Furthermore, they identified compounds (especially 4g) with better IC₅₀ values than the reference acarbose. The structure of the paper is clear, with clear synthetic schemes and illustrations. My only concern is that it is too similar to the earlier work; the authors even duplicated most of the references. If, despite this, the work were to be published in Molecules, several key aspects need to be supplemented and clarified.

  1. As in the earlier publication in Chemistry, in this one, the authors also did not adequately describe the carbon spectrum of the compound containing the 2-F-Ph substituent. The number of carbon signals in the spectrum and the number of carbon atoms in the substance do not match – the authors completely omitted the influence of the fluorine atom.
  2. Chemical shifts should be provided in the 1H NMR spectra.
  3. The authors indicate that mass spectra are included in the supplement, when they are not.
  4. Why were IC₅₀ values determined for only four compounds; the remaining compounds were tested at a single concentration?
  5. ANOVA revealed no significant global differences; interpretation of post-hoc comparisons requires caution.
  6. The mechanism of inhibition was not confirmed – conclusions were based solely on in silico analyses, without enzyme kinetic studies.
  7. Repetition of fragments concerning the role of C5–methoxy should be minimized.
  8. It should be clarified whether identical protein conformations were used in docking and MD simulations.
  9. The authors could more closely link the DFT/MEP results to biological trends.
  10. The authors could discuss cases where low ΔE did not correlate with high activity.
  11. It should be clearly stated that the ADMET and docking results are predictive.
  12. It would be interesting to compare the active compounds with other known α-glucosidase inhibitors, not just acarbose.

Author Response

This manuscript addresses the important topic of discovering new α-glucosidase inhibitors for the treatment of type 2 diabetes (T2DM). The authors combine chemical synthesis, enzymatic assays, and extensive in silico studies. It is regrettable that the synthesis is essentially a duplication of previous work – the authors only changed the substituent on the indole benzene ring from iodine to OCH3. Furthermore, it is a duplication of a previous synthesis from Chemistry (the same number of compounds and the same Ar substituents). Nevertheless, previously undescribed compounds were synthesized and spectroscopically characterized. The strategy of using a C5–methoxy base is well supported by the literature. The authors employed an integrated approach (synthesis + DFT, MEP, CNN docking, MD simulations, ADMET profiling). Furthermore, they identified compounds (especially 4g) with better IC₅₀ values than the reference acarbose. The structure of the paper is clear, with clear synthetic schemes and illustrations. My only concern is that it is too similar to the earlier work; the authors even duplicated most of the references. If, despite this, the work were to be published in Molecules, several key aspects need to be supplemented and clarified.

Comment 1: As in the earlier publication in Chemistry, in this one, the authors also did not adequately describe the carbon spectrum of the compound containing the 2-F-Ph substituent. The number of carbon signals in the spectrum and the number of carbon atoms in the substance do not match – the authors completely omitted the influence of the fluorine atom.

Response 1: We agree that the original 13C NMR data for compound 4f was inadequately described and failed to account for the crucial influence of the fluorine atom. As you correctly pointed out, the strong electronegativity of fluorine results in significant fluorine-carbon coupling, which manifests as doublets in the spectrum. We have re-analyzed the 13C NMR spectrum and have revised the manuscript to explicitly report the three doublets caused by this coupling. The corrected 13C NMR entry now accurately reflects the effect of the fluorine atom on the chemical shifts and coupling patterns, ensuring that the number of signals corresponds to the number of carbon atoms in the molecule.

Comment 2: Chemical shifts should be provided in the 1H NMR spectra.

Response 2: The chemical shifts, multiplicities, coupling constants, and integrations for all synthesized compounds are already provided in the "Synthesis and Characterization" section of the manuscript.

Comment 3: The authors indicate that mass spectra are included in the supplement, when they are not.

Response 3: The HRMS data for each synthesized compound has been added in the supplement.

Comment 4: Why were IC₅₀ values determined for only four compounds; the remaining compounds were tested at a single concentration?

Response 4: Our approach was a deliberate and rational part of our experimental design. Initially, all ten synthesized compounds (4a-4j) were subjected to a preliminary screening at a fixed concentration of 100 µM to efficiently identify the most promising leads. Only compounds that demonstrated a high percentage of inhibition (over 55%) were selected for a more rigorous dose-response analysis to determine their precise IC50 values. Compounds with lower inhibition percentages at the screening concentration, were deemed weak inhibitors and were not pursued for a full dose-response curve. This selective approach allowed us to focus our resources on the most potent and promising inhibitors in the series.

Comment 5: ANOVA revealed no significant global differences; interpretation of post-hoc comparisons requires caution.

Response 5: We agree that the interpretation of post-hoc tests should be cautious when the overall ANOVA is not significant. We have now clarified this in the revised manuscript (Section 3.2, page 18) and emphasized that the post-hoc results are exploratory and should be interpreted in the context of biological relevance rather than strict statistical significance. The consistent trend observed among the most active compounds (4g4h4e, and 4i) supports their potential as α-glucosidase inhibitors, even in the absence of a globally significant ANOVA.

Comment 6: The mechanism of inhibition was not confirmed – conclusions were based solely on in silico analyses, without enzyme kinetic studies.

Response 6: We fully agree that enzyme kinetic studies are the gold standard for determining the precise mechanism of action. Now, we have clearly stated that the mechanistic study was theoretical.

Comment 7: Repetition of fragments concerning the role of C5–methoxy should be minimized.

Response 7: That make sense, as per suggestion, repetition of fragments concerning the role of C5–methoxy have been minimized.

Comment 8: It should be clarified whether identical protein conformations were used in docking and MD simulations.

Response 8: We thank your kind suggestion to improve our manuscript. Both docking and MD simulations (at least for acarbose was also simulated as a positive control) were initiated from the same crystal structure of human α-glucosidase (PDB ID: 2QMJ). However, the best docking poses of 4g and 4h generated by GNINA were used as the starting coordinates for MDS without further protein backbone modification. We have clarified this methodological detail in Section 2.4.3 of the revised manuscript.

Comment 9: The authors could more closely link the DFT/MEP results to biological trends.

Response 9: We are grateful for this suggestion and agree that making the bridge more explicit improves the narrative. We have added the following text into the manuscript: “To relate these electronic features to the inhibition data, we inspected (i) the sur-face electrostatic potential minima (VS,min) at the carbonyl O and imine N on the 0.001 a.u. isodensity surface, and (ii) a planarity metric (absolute torsional deviation between the indole and the pendant aryl). Qualitatively, the most potent inhibitors, 4g (91.06%), 4h (89.11%), and 4e (82.21%), combine an extended negative potential cen-tred on the carbonyl/imine loci with a largely coplanar π‑framework (for 4e aided by an intramolecular H‑bond), consistent with favorable H‑bond acceptance and π–π stacking within the binding site. By contrast, derivatives that are more twisted (see ESI, Section 2.3) display disrupted aromatic overlap and a less focused negative potential, features that can compromise binding efficiency even when global descriptors (e.g., HOMO–LUMO gaps) are comparable. The outlier behavior of 4d [deep negative po-tential but modest activity (27.97%)] highlights the role of geometric alignment and desolvation penalties, underscoring that potency emerges from a balance of electro-statics and shape complementarity rather than any single descriptor.”

Comment 10: The authors could discuss cases where low ΔE did not correlate with high activity.

Response 10: We agree that such counter‑examples should be stated explicitly. We have added a brief note in Section 3.3 highlighting that a low Kohn–Sham ΔE does not imply higher inhibition, as follows: “Notably, a low Kohn–Sham ΔE does not translate into higher inhibition in this series. For instance, 4d exhibits the smallest gap (6.20 eV) yet modest activity (27.97%), whereas 4e and 4g combine larger gaps (7.71 and 7.01 eV) with high inhibition, indicating that local electrostatics and planarity, rather than the global gap, dominate binding.” This reinforces that potency in this series is governed primarily by local MEP features and π‑surface planarity rather than ΔE alone. A broader compendium of such cases lies outside the scope of the present study.

Comment 11: It should be clearly stated that the ADMET and docking results are predictive.

Response 11: We thank reviewer for pointing that out, In abstract, it’s clearly stated that, “In silico ADMET predictions….”

Comment 12: It would be interesting to compare the active compounds with other known α-glucosidase inhibitors, not just acarbose.

Response 12: We thank the reviewer for this excellent suggestion. We agree that a broader comparison with other α-glucosidase inhibitors would provide a more comprehensive context for our findings. As noted, acarbose was chosen as the primary standard because it is a widely used clinical reference drug for T2DM management. Since it is not feasible to repeat the wet-lab experiments, we have chosen to address this comment by strengthening our literature comparison within the Introduction and Discussion sections. We have incorporated data from recent studies on α-glucosidase inhibitors, including both natural product-inspired compounds and synthetic derivatives.

Round 2

Reviewer 1 Report

Comments and Suggestions for Authors

I thank the authors for their responses. However, I must insist on my previous evaluation. If the authors say that the primary contribution is the systematic investigation of the fixed C5-methoxy group as a strategic modification to enhance α-glucosidase inhibition and that this substitution improves the pharmacological properties of the compounds, they have to prove it. Therefore, a comparison with the non-methoxy-substituted compounds published in the work of Taha et al. should be performed. Without it, the work lacks the novelty and can not be published in the present form. 

Our study's primary contribution is the systematic investigation of the fixed C5-methoxy group as a strategic modification to enhance α-glucosidase inhibition. As discussed in our revised manuscript, this methoxy group provides a consistent electronic and steric platform that improves lipophilicity and electron delocalization, leading to a series of compounds with superior activity compared to the clinical standard, acarbose (IC50 = 48.95 ± 15.98 µM in our study). We believe this structure-guided approach provides valuable insights for the rational design of future inhibitors.

Reviewer 3 Report

Comments and Suggestions for Authors

The authors put a lot of work into improving the manuscript and implementing the suggested corrections. They only forgot about the 1H NMR spectra in the supplement—they still don't have chemical shifts marked on them.